# An Attack on Zawadzki’s Quantum Authentication Scheme

**DOI:** 10.3390/e23040389

**Published:** 2021-03-25

**Authors:** Carlos E. González-Guillén, María Isabel González Vasco, Floyd Johnson, Ángel L. Pérez del Pozo

**Affiliations:** 1Departamento de Matemática Aplicada a la Ingeniería Industrial, Universidad Politécnica de Madrid, 28040 Madrid, Spain; carlos.gguillen@upm.es; 2MACIMTE, Universidad Rey Juan Carlos, 28933 Madrid, Spain; mariaisabel.vasco@urjc.es (M.I.G.V.); angel.perez@urjc.es (Á.L.P.d.P.); 3Department of Mathmatical Sciences, Florida Atlantic University, Boca Raton, FL 33431, USA

**Keywords:** quantum identity authentication, private equality tests, conclusive exclusion

## Abstract

Identification schemes are interactive cryptographic protocols typically involving two parties, a prover, who wants to provide evidence of their identity and a verifier, who checks the provided evidence and decides whether or not it comes from the intended prover. Given the growing interest in quantum computation, it is indeed desirable to have explicit designs for achieving user identification through quantum resources. In this paper, we comment on a recent proposal for quantum identity authentication from Zawadzki. We discuss the applicability of the theoretical impossibility results from Lo, Colbeck and Buhrman et al. and formally prove that the protocol must necessarily be insecure. Moreover, to better illustrate our insecurity claim, we present an attack on Zawadzki’s protocol and show that by using a simple strategy an adversary may indeed obtain relevant information on the shared identification secret. Specifically, through the use of the principal of conclusive exclusion on quantum measurements, our attack geometrically reduces the key space resulting in the claimed logarithmic security being reduced effectively by a factor of two after only three verification attempts.

## 1. Introduction

One of the major goals of cryptography is authentication in different flavours, namely, providing guarantees that certain interaction is actually involving some specific parties from a designated presumed set of users. In the two party scenario, cryptographic constructions towards this goal are called identity authentication schemes, and have been extensively studied in classical cryptography [1,2]. Classically, there are different ways of defining so-called identification schemes, for mutual authentication of peers, mainly depending on whether the involved parties share some secret information (such as a password) or should rely on different (often certified) keys provided by a trusted third party. The advent of quantum computers may suggest the end for many of these protocols however.

Since Wiesner proposed using quantum mechanics in cryptography in the 1970s, multiple directions using this concept have undergone serious research. One major role quantum mechanics has played in cryptography is the development of quantum key distribution (QKD) where two parties can securely share a one time pad using quantum mechanics, for example, the seminal protocol BB84 [3]. Among protocols providing entity authentication and strictly quantum in nature, some of them, such as those in [4,5,6], are based on entanglement, while more recently [7,8] do not rely on entanglement but rather propose to obtain identity authentication evidence from only the common knowledge of a shared secret. These approaches are known as quantum identity authentication (QIA) protocols (see also the related papers [9,10,11,12,13,14]). Due to the existence of quantum protocols such as BB84 that do not rely on entanglement it would be more appealing to not rely on entanglement for entity authentication purposes.

The QIA constructions in which authentication is intended from the common knowledge of a shared secret, often called QIA schemes (or just quantum identification schemes), are closely related to protocols for quantum equality tests and quantum private comparison. All these constructions are concrete examples of two-party computations with asymmetric output, i.e., allowing only one of the two parties involved to learn the result of a computation on two private inputs. Without imposing restrictions on an adversary it was shown by Lo [15], Colbeck [16] and Buhrman et al. [17] that these kind of constructions are impossible, even in a quantum setting. As a consequence, constructions for generic unrestricted adversaries in the quantum setting are doomed to failure.

While there are many things in common in the frameworks for developing QKD protocols and identification schemes built as private comparison tests, we make note of the following key differences in cryptographic considerations. Most QKD setups involve an authenticated classical channel, thus the recipients may safely compare check bits to see if there is an unintended observer. This however may not be the case in an authentication scheme (like the one considered in this paper), so there may be no way for the legitimate parties to determine if an eavesdropper is present. Thus, if the states obtained by the authenticating party are not as expected, the authentication fails without the users knowing if it is due to an adversarial presence or an attempted impersonator. For this reason the traditional so called intercept-and-resend attack is completely irrelevant for authentication as the adversary is always capable of sending messages as if coming from Alice or Bob, though without the correct private value the protocol is overwhelming likely to fail. The closest equivalent constraint is that the authenticating party may only make a single measurement on a qubit before the state collapses. This constraint bars the adversary from making many measurements on the same state in order to fully receive the private value. This however does not exclude the possibility that many different calls of the authentication protocol are made. Unlike key distribution protocols, where after a failure the key is discarded, both classical and quantum authentication protocols must be secure after being run multiple times with the same shared secret though with different random inputs [1]. We make special note here that the objectives of QKD and QIA schemes are very different. With this in mind readers should be cautious to apply the results of this work to any current or future scheme if and only if its objectives and methods fall within certain parameters.

### 1.1. Our Contribution

Recently, an original work about authentication without entanglement by Hong et al. in [8] was improved by Zawadzki using tools from classical cryptography in [7]. In Zawadzki’s protocol, there are two parties, Alice and Bob, who share a common secret bitstring *k*. In order to achieve entity authentication from Alice to Bob, they run a non-interactive protocol in which Alice first computes a hash value $h_{a},$ which depends on *k* and a random nonce r; then Alice sends *r* to Bob so he can reproduce the computation obtaining a hash value $h_{b}$ (which must equal $h_{a}$ if there is no adversarial interference). Next Alice sequentially sends quantum states to Bob, which she prepares as a function of consecutive pairs of bits of *h*. At reception, Bob measures these states choosing each time a basis which depends on the value *h*. If all measures’ outcomes are the expected ones, Bob concludes that the other party must know *k* and, therefore, identifies it as being Alice.

Our theoretical analysis of the protocol shows its insecurity, but in a non-constructive way (e.g., it does not help finding a concrete successful adversarial strategy). However, we are in addition able to show an explicit attack against the protocol, based on conclusive exclusion on quantum measurements, which we describe in Section 4. There we analyze in detail how the attack halves the size of the key space after only three verification attempts.

Note that, when analyzing Zawadski’s protocol, we deal only with its theoretical design. Both the impossibility results we invoke and our attack do not take advantage of physical aspects, such as distance or timing, they hold independently of the implementation. It is indeed interesting to study in depth how identification protocols could be practically deployed in the real world, and what weaknesses could be exploited, but this is beyond the scope of this work. These physical issues, present in attacks against QKD, such as, for example, time-shift attacks [18], phase-remapping attacks [19] or synchronization attacks [20], would also naturally arise for quantum identification protocols.

Finally, we discuss the applicability of the impossibility results and the explicit attack to other QIA protocols, such as [4,5,8,21,22,23,24]. For instance, we point out that the protocol from Hong et al. [8], in which Zawadzki’s protocol is based upon, is vulnerable to the same attack we describe against the latter. On the other hand, the rest of the protocols cited, for different reasons discussed later, are neither affected by the impossibility results nor vulnerable to our attack.

The main contribution that arises from this work is that our theoretical analysis evidences an implication of the proven impossibility of identification schemes, such as Zawadzki’s design. Thus, we stress that fundamental changes in the original proposal, beyond preventing our particular attack, would be needed in order to derive a secure identification scheme.

### 1.2. Paper Roadmap

We start this contribution by summarizing in Section 2 the impossibility results from Lo [15], Colbeck [16] and Buhrman et al. [17], concerning generic quantum two party protocols. Further, we present and discuss the Zawadzki protocol in Section 3, evidencing it actually fits the framework considered in the impossibility results from Section 2, and thus concluding it must necessarily be insecure. Moreover, we outline a simple explicit attack which we describe in Section 4. Finally we discuss how other QIA protocols are affected by our results in Section 5 and provide some conclusions in Section 6.

## 2. Quantum Equality Tests Are Impossible

A one sided equality test is a cryptographic protocol in which one party, Alice, convinces another party, Bob, that they share a common key by revealing nothing to them but equality (or inequality) of their inputs. Formally we define a key space *K* and a function $F:K^{2}\to \left\{0,1\right\}$ which checks for equality. Let i∈K be Alice’s key and j∈K be Bob’s key. The goals of a one sided equality test are as follows:(1)F(i,j)=1 if and only if i=j.(2)Alice learns nothing about *j* nor about F(i,j).(3)Bob learns F(i,j) with certainty. If F(i,j)=0 then Bob learns nothing about *i* besides that i≠j.

The above is a specific case of a one-sided two-party secure computation protocol as described in [15], as only one side, Bob, learns the output of the computation. In this work, a very general result is proven indicating that any protocol realising a one-sided two-party secure computation task is impossible, even in a quantum setting. In particular, Lo shows in [15] that if a protocol satisfies (1) and (2) then Bob can know the output of F(i,j) for any *j*. Furthermore, a one sided equality test with some small relaxations on points (1) and (3) is also proven impossible. Hence, any one-sided QIA protocol which validates identities using equality tests by use of quantum mechanics is impossible without imposing restrictions on the adversary.

Note that the above argument says nothing about protocols with built in adversarial assumptions such as those presented in [25,26]. Further, note that many of QIA schemes in the literature include a final round where Bob accepts or rejects, which makes Alice aware of the success or failure of the protocol. Indeed, those schemes can be straightforwardly turned into one-sided equality tests by suppressing Bob’s final message announcing the result. Hence, they are clearly insecure against a dishonest Bob. However, note that if any such protocol can be modified so that Alice may obtain information on the identification output at some point before the last protocol round, it is unclear how Lo’s impossibility result would apply. However, if they are built upon equality tests we can get impossibility from another well know result by Buhrman el al. [17]. Certainly, two-sided QIA schemes, in which both Alice and Bob learn the result of the protocol, are a particular case of two-sided two-party computations. It is shown in [17] that a correct quantum protocol for a classical two-sided two-party computation that is secure against one of the parties is completely insecure against the other. For equality tests, if one of the parties, say Alice, learns nothing else than F(i,j), the other party, Bob, will indeed be able to compute F(i,j) for all possible inputs *j*. Thus, any two-sided QIA protocol which validates identities using equality tests is also impossible without imposing further restrictions on the adversary.

Both total insecurity results are valid for protocols that compute a deterministic function *F*, and admit relaxed versions for computations that implement approximate versions of *F*. For a non-deterministic function *F*, Colbeck [16] showed that in a correct one-sided or two-sided two-party computation for *F*, one of the parties can always access more information about the other party’s input than it is supposed to, where the analysis is only done quantitatively for dychotomic values of *i*,*j*, and extended trivially to the general case, yielding a qualitative more than a quantitative result.

## 3. Insecurity of Zawadzki’s QIA Protocol


In this section, we outline the protocol proposed in [7] and show that it must be insecure on Alice’s side by the results discussed in Section 2. Moreover, we consider minor changes to the protocol to evidence that making it more “in line” with classical authentication does not help, as the protocol remains insecure. Indeed, the changes introduced do not fundamentally alter the protocol, namely both the changed and unchanged protocols allow for the attack we outline in Section 4 to provide information leakage.

The protocol proposed in [7] can be described as follows: suppose Alice and Bob have keys ka and kb, respectively, and agree on some universal hash function (universal hash functions are to be understood as families H of functions providing a nice collision-resistance property, i.e., given inputs x≠y, the probability of h(x)=h(y) can be proven negligible if *h* is chosen at random from H (see [27]). In an abuse of notation, is it typical to treat them as individual functions, as we do above) H:{0,1}N→{0,1}2d. Bob wishes to verify that kb=ka without leaking any information about kb or ka. Alice randomly generates a nonce ra from a designated domain and calculates the value ha=H(ra||ka). Alice sends Bob ra. Bob receives rb (which in principle should be equal to ra) then calculates the value hb=H(rb||kb). Note that if ka=kb and the nonces are received as constructed, then ha=hb. Alice then acts on pairs of bits in ha with an embedding function Q:{0,1}2→ℂ2. This function *Q* uses the first of the two binary values to determine the measurement basis (horizontal/vertical or diagonal/antidiagonal) and the second to determine the specific qubit in {|0〉,|1〉,|+〉,|−〉}. More precisely, Q(0,0)=|0〉, Q(0,1)=|1〉, Q(1,0)=|+〉 and Q(1,1)=|−〉. Applying *Q* to the pairs of bits in ha Alice prepares and sends *d* qubits to Bob over the quantum channel one by one with a constant speed known to Bob.

Using the first bit of each pair Bob decides in which base he measures the quantum states and insures he obtains the correct qubit according to the second bit of the pair. If the loss of qubits is very high or the rate of bits measured by Bob that disagree with the even bits of $h_{b}$ is over a certain threshold then Bob rejects Alice’s challenge. Otherwise he accepts her challenge. See Figure 1 for a schematic overview of the protocol.

For the sake of simplicity we restrict the security analysis to the case where there are no losses in the communication and the bit error rate is set to 0.

The Zawadzki protocol is claimed to be leakage resistant when considering an adversary measuring in a random basis. The reasoning behind this is that unless an adversary, Eve, correctly guesses the correct basis for each round, she will obtain different values for at least one of the bits of the hash. Now suppose an adversary is capable of computing preimages of hash functions through brute force with unbounded classical computational power or through dictionary attacks with unbounded classical memory. In this case it is unlikely that there will exist a $k_{e}\in K$ such that $H(r_{e}||k_{e})$ matches what Eve measured. In the event there does exist such a $k_{e}$ then with overwhelming probability $k_{e}\neq k_{a}=k_{b}$ and Eve will not be able to falsify authentication of Alice or Bob.

Unfortunately, Zawadzki’s protocol implemets a two-sided equality test (one-sided if the last accept/reject round is omitted) for the secrets, with a relaxation on the correctness, that is, the condition F(i,j)=1 if and only if i=j (in this case *i* is $h_{a}$ and *j* is $h_{b}$). Suppose, for the sake of reasoning, that the protocol were a correct two-sided equality test, then all the results summarized in Section 2 apply and the protocol has necessary leakage. As Bob is sending nothing but the final bit, we know that nothing can possibly leak from $h_{b}$. Thus, any potential leakage comes from $h_{a}$ and in fact it is completely leaked. Although Eve may not be able to determine any exact bit of $k_{a}$, due to collisions of the hash function, she may drastically reduce the number of possible options for $k_{a}$ to those *k* such that $h_{a}=H(r_{a}\left||k\right)$ and hence construct a proper subset of *K* such that the true value for $k_{a}$ is contained in this subset.

However, Zawadzki’s protocol is not perfectly correct. Whenever Alice and Bob secrets, $h_{a}$ and $h_{b}$, differ in the measurement bits (the ones associated to the measurements basis), there is some probability of the computation returning value 1 and thus Bob accepting Alice’s input as valid. This probability is exponentially small in the number of different measurement bits between $h_{a}$ and $h_{b}$, that is, for a large majority of the cases this probability is very small. Thus, the reasoning made in the approximate case of the relaxation of the correctness in the one-sided case in [15] can be applied to Zawadzki’s protocol (without the last round) in these cases. That is, when Bob chooses a secret that differs in many measurement bits from Alice’s secret, what will happen for a random choice of the secret, he will be able not only to compute with some probability (close to 1) the equality test for $\left(h_{a},h_{b}\right)$, but to compute the equality test with some different probabilities (close to 1) for every $\left(h_{a},h_{b}^{\prime }\right)$ such that the output of the computation has large probability of being the value of the equality test. Thus, he will obtain partial (but close to full) information about many different secrets at the same time.

The approximate version of the result of Buhrman et al. [17] does not straightforwardly say anything in this case as their notion of approximate correctness requires that the function *F* should be computed correctly for every input with probability close to 1. Whereas in Zawadzki’s proposal the pairs of secrets $\left(h_{a},h_{b}\right)$ that only differ in one of the measurement bits has probability of computing correctly the equality test equals 1/2. However, it may be possible to give a version of the result of Buhrman et al. with a different notion of approximate correctness.

Finally, the result of Colbeck does apply when considering the non deterministic function *F* to be the actual computation of the secrets $h_{a}$ and $h_{b}$ implemented by the protocol. Thus, the function implemented by the protocol is not secure and a dishonest Bob could learn information about the implemented function for more than one secret $h_{b}$ at a time, acquiring more information than following the protocol honestly.

Next we analyze what happens if some minor changes are done to make the protocol more in line with classical authentication schemes. Unfortunately, we conclude that these changes do not fundamentally modify the protocol and as will be clear the previous reasoning still holds. Moreover, both the changed and unchanged protocols still allow for the particular attack outlined in Section 4 to provide information leakage by allowing an adversary to learn about many $h_{a}$ simultaneously as predicted by the results of Lo and Colbeck.

Changes made to the protocol are as follows: (1) Bob generates *r* and *H*, this is done to thwart a simple attack discussed later; (2) the hash function changes between trials, this has no impact on the security of the protocol due to the public nature of the hash in both instances; and finally (3) here we assume for simplicity that Alice and Bob obtain the same nonce *r* with certainty, using classical error correction techniques one can be relatively certain both parties obtain the same nonce. See Figure 2 below for a schematic overview of the modified protocol.

The reason we force Bob to generate the randomness instead of Alice is that an adversary with unbounded quantum memory may impersonate Bob but not make a measurement. Suppose an adversary does not know the key but requests Alice to identify herself. If Alice generates and sends r,H with the string of states $|\varphi_{i}\rangle$ then the adversary may record r,H and hold in memory, but not measure, the qubits. At a later time an honest participant may ask the adversary to identify themselves, in this case the adversary may send r,H and the qubits in memory. Thus, the adversary correctly forges an authentication. Note that as we have presented the algorithm an adversary may still make this impersonation by waiting between Alice and Bob then passing the information between the two. The difference is that as long as Bob generates the nonce then this attack must only be done while Alice and Bob are both online, whereas if Alice generates and sends the nonce then an adversary may hold the states for as long as is technologically feasible.

Unfortunately, the changes introduced do not alter the validity of the impossibility results discussed before. This updated version is still a two-sided equality test (one-sided if the last accept/reject round is omitted) for the secrets with a relaxation on the correctness, as no changes have been introduced after the generation of the secrets.

Thus, both the original and the modified protocols have necessary leakage and due to the non-interactive nature of Bob we know that $k_{b}$ has no leakage, thus we know there must exist some leakage on $k_{a}$. Although Eve may not be able to determine any exact bit of $k_{a}$ she may drastically reduce the number of possible options for $k_{a}$ and hence construct a proper subset of *K* such that the true value for $k_{a}$ is contained in this subset. An attack exemplifying this phenomenon is described in the next section.

## 4. A Key Space Reduction Attack on Zawadzki’s Protocol

Before discussing the specific attack, let *B* be a set of orthogonal bases in $C^{2}$ and consider the following fact. If a quantum state is prepared in a basis b∈B with value v∈{0,1}, then an adversary may always remove one possible combination of *b* and *v* with a single measurement. Upon measuring in basis $b^{\prime }\in B$ an adversary obtains $v^{\prime }\in \left\{0,1\right\}$. The adversary is then certain the original pair (b,v) was not $\left(b^{\prime },1⨁v^{\prime }\right)$, as when measured in the basis *b* the qubit prepared by *b* and *v* will yield *v* with certainty. Note that the adversary cannot say with certainty how the qubit was prepared, but he can always remove one possible option. This is an example of conclusive exclusion discussed in [28] in the case of two measurement bases.

Suppose now that instead of sampling at random for *b* and *v*, the qubit is prepared using a private key k∈K and a set of public parameters *p*, namely b=b(k,p) and v=v(k,p). An adversary once again measures in basis $b^{\prime }\in B$ (chosen or taken at random) to obtain $v^{\prime }\in \left\{0,1\right\}$, they may then determine a basis/value pair in which the qubit was not prepared. Because the adversary is assumed to be computationally unbounded they may then compute $b(\hat{k},p)$ and $v(\hat{k},p)$ for all $\hat{k}\in K$. Whenever these computations output the impossible pair $k^{\prime },v^{\prime }$ the adversary becomes aware that $\hat{k}\neq k$, hence reducing the key space. The extent to which the key space is reduced depends on the number of basis in *B*. If the distribution of basis choices in *B* is low entropy the attack may be accomplished as described while if *B* is high entropy then a probabilistic version decreases the space of likely keys. The assumption that the adversary is computationally unbounded may be lifted if *k* is low entropy (for he can then indeed test all possible values for *k*—given there are only a polynomial set of candidates), however assuming a computationally bounded adversary immediately removes unconditional security as an end goal.

Let us now apply this key space reduction to the QIA protocol proposed in [7], in this case the private key is *k* and the public parameters are *r* and *H*. Suppose an Eve has no a priori knowledge of the key except its existence in *K*. After receiving *r* and *H* over the classical channel she measures all qubits $|\varphi_{i}\rangle$ received from Alice in the horizontal/vertical basis and records the outputs as *M*. In the case where Eve is utilizing man-in-the-middle, she is done. If she is impersonating Bob, she accepts or rejects the protocol.

After the protocol finishes the adversary may then compute $h_{\hat{k}}=H(r||\hat{k})$ for all $\hat{k}\in K$. Suppose the first qubit Eve measured in *M* was |0〉. She now examines the first two bits of each $h_{\hat{k}}$, those that begin 00, 10, or 11 are all possible of obtaining the qubit |0〉 after measurement. The first of these three tuples will yield |0〉 with certainty and the later two with a probability of 0.5. The final tuple 01 however is not possible as that would imply that the qubit started in the state |1〉 and measured in |0〉. Thus, Eve knows that any $\hat{k}$ such that $h_{\hat{k}}$ begins 01 is not the key. The hash function is assumed to be independent and identically distributed so this removes approximately $\frac{1}{4}$ of all possible keys. Repeat this process for all qubits. After completion of all hash and check operations the adversary has obtained a subset of the key space which contains the key, hence causing information leakage. Specifically, the adversary knows the key is in subset *S* defined by
$$S=\{s\in K:h_{s_{2i}}=M_{i}\;\mathrm{and}\;h_{s_{2i-1}}=0\;\forall i\leq d\}.$$

Note that the true key k∈S and $\left|S\right|\approx {\left(\frac{3}{4}\right)}^{d}\left|K\right|$.

After running this attack on a single attempted authentication the proposed ideal (brute force) security of $2^{2d}=2^{N}$ drops to $3^{d}=2^{log_{2}\left(3\right)/2N}\approx 2^{0.792N}$. Recall that authentication protocols must remain secure given many attempts. Thus, an adversary is allowed to receive multiple authentication attempts, possibly claiming that the received hash of the shared secret is denied due to interference from a third party. The logarithm of the security parameter drops geometrically at a rate of $\frac{log_{2}\left(3\right)}{2}\approx 0.792$ after every authentication the adversary receives, meaning that once an adversary obtains the third authentication (all with different random values or even different hash functions) the brute force security has been reduced to brute force on a string of half the length. This trend continues with every authentication attempt.

## 5. Other QIA Protocols

It is worth pointing out that the attack described in Section 4 also applies to the protocol by Hong et al. [8], which Zawadzki [7] modifies. In more detail, the protocol in [8] is similar to Zawadzki’s, but does not use a hash function. Instead, whenever Alice transmits the qubits sequentially and, before sending each qubit, she randomly decides if she is going to use security mode or authentication mode. In the first case, she sends a decoy state while in the second one, a qubit encoding two bits of the authentication string is sent, similarly to [7]. After Bob’s reception, Alice announces which mode she just has used. Therefore an adversary using the same strategy described in our attack in Section 4 and collecting the information obtained whenever Alice announces authentication mode, will be able to shrink the size of the key space in the same way we have previously stated.

On the other hand, other quantum identification protocols proposed in the literature are not vulnerable to our attack neither contradict the impossibility results mentioned in Section 2. For instance, some of them [4,5,21] are aided by the presence of a trusted third party, therefore not being real two-party protocols. Another type of protocols, such as [22,23,24], make use of an entangled quantum state shared between both parties. In [22], the users, in addition, share a bitstring used as a password; both parties measures their part of the entangled state to produce a one time key that one of the users XORs with the password and sends the result to the other who checks for consistency. The downside of this approach is that to repeat the identification process the parties need to be provided again with new entangled states. In [23,24], the users do not share any classical secret, they just use the entangled state to identify themselves.

## 6. Conclusions

The protocol given by Zawadzki in [7] may be secure against hash preimage attacks when attempting to find an exact match; however, when considering impossible results from quantum measurements we see some hashed key values are not possible. Proverbially, the forest may be secure but each of the trees reveals enough information to reconstruct the possible forests. By eliminating approximately one quarter of the key options from each qubit we see that by measuring all the individual qubits in a random basis does in fact reveal a great deal about the key. This attack does not concern quantum memory but rather relies heavily on classical computational power. Hence, unlike [25,26] where the authors consider a bounded quantum storage model, the only way to make this protocol secure without greatly changing its construction is to constrict adversarial computational power.

No solution is presented to the problem outlined in this paper. The reason for this is that any solution presented which does not impose more fundamental restrictions such as limited quantum memory or polynomial time restriction will inevitably fail due to the results of Lo [15], Colbeck [16] and Buhrman et al. [17]. Regardless of the restriction imposed, implementation of this and any other “prepare and measure” authentication scheme must find a way to contend with key space reductions posed by conclusive exclusion.

## Figures and Tables

**Figure 1 entropy-23-00389-f001:**
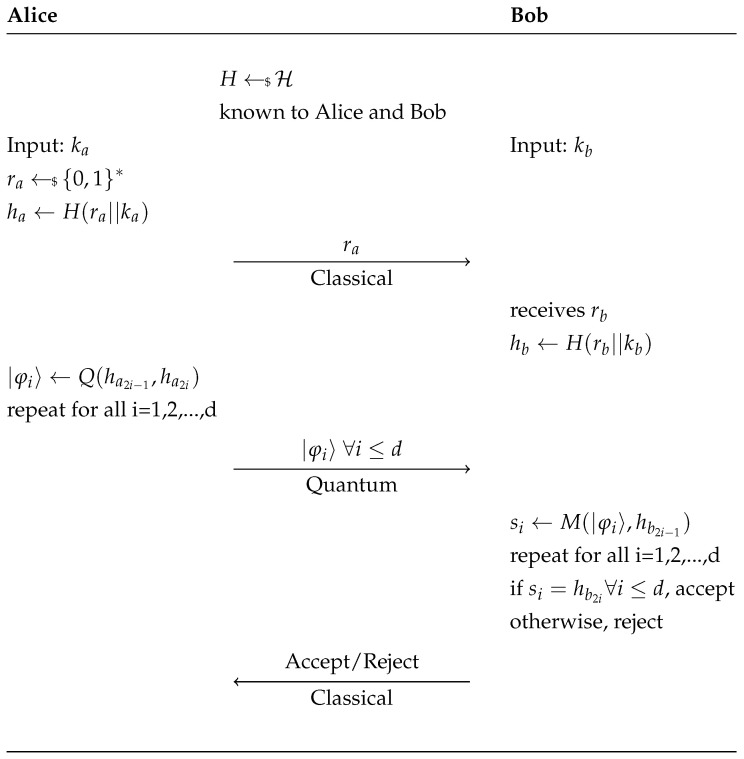
The protocol presented in [7].

**Figure 2 entropy-23-00389-f002:**
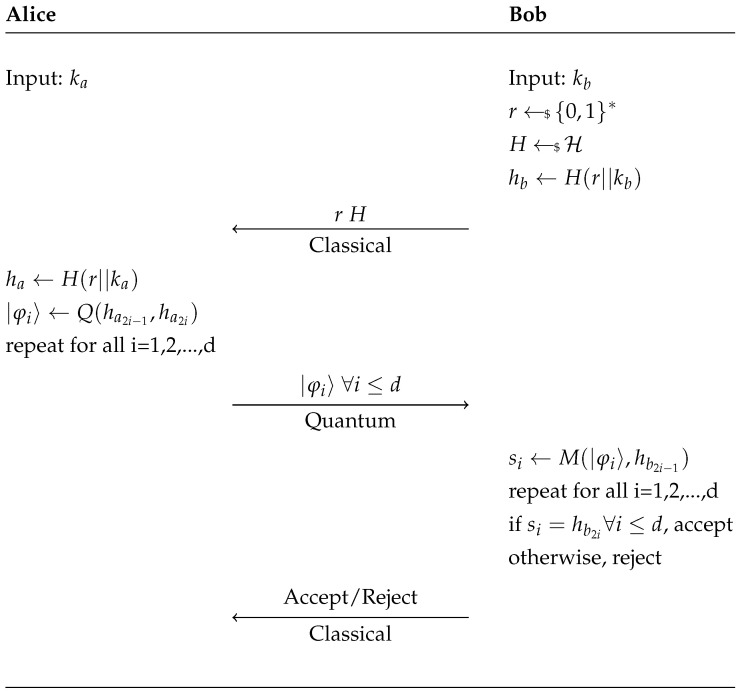
Modified protocol.

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
