# Peer review of "An Attack on Zawadzki’s Quantum Authentication Scheme"

_entropy, 2021, doi:10.3390/e23040389_

Round 1

Reviewer 1 Report

In this work the authors propose an attack against the authentication protocol proposed recently by P. Zawadzki (Ref. 1). There are many serious problems with the present manuscript, and thus I cannot recommend its publication in Entropy.  

(a) The authors have confused entity authentication with data origin (message) authentication. The protocol they discuss is an entity authentication protocol, and even if it were correct and secure, it does not offer anything in the framework of QKD. This is because secure QKD requires secure data origin authentication, and not entity authentication. The latter does not necessarily imply the former, and vice versa. This is why they constitute different research topics in cryptography.

(b) There is no clear connection of the scheme of Zawadzki to Refs. [2-4]. I do believe that the work Zawadzki is useless and insecure in a practical setting, under realistic conditions. In the present manuscript the authors do not address these obvious vulnerabilities, while their discussion is very confusing. If the authors wish to point out these vulnerabilities, it can be done in a simple and concise way, within three pages at most. Their work should be sent as Comment to the work of Zawadzki, so that other scientists who are interested in the same topic become aware of these vulnerabilities.

(c) The work is not clearly written, as it suffers from many grammar and syntax problems.

Author Response

a) The reviewer makes the correct assessment about the difference between entity authentication and message authentication.  However, it is not quite correct that entity authentication is irrelevant for QKD. While this may be true for some concrete scenarios, most QKD protocols require an authenticated classical channel (this is the case, for instance, for BB84). This authenticated channel is actually providing entity authentication and not data origin authentication. For example, in BB84 participants must be able to send polarization states (horizontal/vertical or diagonal/antidiagonal) after sending or receiving the actual qubits for establishing a shared key, this kind of interactive design completely rules out the possibility of data origin authentication.   b) The reviewer claims the work should be submitted as a short comment on Zawadzki’s scheme, which is a fair comment.  However, we believe as we mention in the article that the root issue of the problems outlined through discussion is larger than that of the Zawadzki protocol. As outlined in citations 2-4 private comparison tests frequently used in password authenticated key exchanges cannot be se-cure in a quantum setting without imposing further restrictions on the adversary. We believe this is of interest to the broader quantum cryptography community beyond those working directly with the Zawadzki protocol. As the reviewer points out, the practical issues making Zawadzki’s protocol useless/insecure are surely simpler to state, yet we demonstrate that beyond that there are fundamental theoretical rea-sons to disregard Zawadzki’s proposal.  With respect to the connection between Zawadzki’s protocol and references [2] to [4], we actually have explained them in depth in Section3. To clarify this connection from the very beginning of the paper, following the reviewer’s advice, we have added a new paragraph to the our contributions subsection. c) We have carefully revised the writing and we are fairly confident it does not have meaningful mistakes

Reviewer 2 Report

This manuscript comments on a recent proposal for quantum identity authentication from Zawadzki [1]. The authors discuss the applicability of the theoretical impossibility results from Lo [2], Colbeck [3], and Buhrman et al. [4] and formally prove that the protocol must necessarily be insecure. Moreover, through the use of the principle of conclusive exclusion on quantum measurements, this attack geometrically reduces the key space resulting in the claimed logarithmic security being reduced effectively by a factor of two after only 15 three verification attempts. As far as I known, both the attack strategy and the security analysis are correct. Indeed, I found the idea is original and interesting. Thus, I can recommend this manuscript for publication in Entropy.

Author Response

 We are glad the reviewer found our work interesting and thank him/her for his/her time and effort.

Reviewer 3 Report

Interesting article. The topic is relevant. Section 2 does not provide a detailed explanation of the attack, and analogs are not considered sufficiently. Does distance matter? What about other protocols? Now the systems of the new generation (Сlavis3) and new protocols (COW for ex.) .. how with them?
Nevertheless the study is interesting and deserves to be continued ...

I have no access to the borrowing system, so I cannot judge plagiarism. I recommend adding sources ... for example

A.P. Pljonkin. Vulnerability of the Synchronization Process in the Quantum Key Distribution System. International Journal of Cloud Applications and Computing. V. 9, I. 1, 2019. DOI: 10.4018/IJCAC.2019010104.

Author Response

a)  We have added a paragraph in the our contribution subsection explaining that, in this work, we are not concerned with physical issues, as both the impossibility results we invoke and our attack exploit deficiencies in the theoretical design of the protocol. In this paragraph we also acknowledge the interest of these physical issues regarding quantum identification protocols. It would be an interesting topic for follow up work in this area and we thank the reviewer for making us aware of it. With respect to our particular attack against Zawadzki's protocol, it is explained with detail in Section 4.

b) Pertaining to the suggested source,  In the same paragraph mentioned above, we have also added a quote to this work to exemplify how this kind of attacks are relevant to QKD and could also appear against quantum identification protocols. We thank the reviewer for the pointer. We have also added two other new references pointing to other attacks against QKD protocols which exploits physical issues.

Round 2

Reviewer 1 Report

I disagree with the authors. Secure QKD requires data origin authentication (also known as message authentication) and not entity authentication as the authors claim. To prevent man-in-the-middle attack, one has to ensure the origin of the data  that are transmitted during the public discussion of a QKD protocol. Entity authentication does not tell us anything about the origin of the data that will be exchanged after a successful entity authentication, as one can intercept the transmission at any time. Some of the protocols that are used in practical QKD are typical MACs (message-authentication codes) and rely on universal hash functions, which offer unconditional security. 

For instance, the authors may look at the following sources: 

https://www.etsi.org/deliver/etsi_gs/qkd/001_099/002/01.01.01_60/gs_qkd002v010101p.pdf

https://www.researchgate.net/publication/271369321_Authentication_in_Quantum_Key_Distribution_Security_Proof_and_Universal_Hash_Functions

https://marketing.idquantique.com/acton/attachment/11868/f-020d/1/-/-/-/-/Understanding%20Quantum%20Cryptography_White%20Paper.pdf

as well as on the following work for differences between classical and quantum message authentication codes

https://www.mdpi.com/2410-387X/4/4/31

Of course one may also use classical digital signatures, which provide simultaneously data origin authentication and entity authentication. However, the main aim is still message authentication and not entity authentication. The latter comes as a bonus, and it is not necessary for the security of the QKD. Moreover, it has to be emphasised that contrary to universal hash functions, digital signatures provide computational security and this will affect the security of the entire QKD.  

To summarize, my opinion is that the authors discuss entity authentication,  and their work does not have any impact whatsoever on QKD, which requires message authentication. The work should be decorrelated from QKD, to avoid further misunderstandings.  If, however, the authors believe that the protocol they discuss also achieves message authentication, then it is already known that quantum schemes do not offer anything relative to classical unconditionally secure MACs typically used in QKD (see paper by Fischlin and co-workers), and thus the work is of no importance.  I have given references to support my claims, and I could mention hundreds of papers. The  authors should look carefully at these sources and the references therein, so that to clarify this issue. For the time being, the manuscript remains as confusing and misleading as before. As I wrote in my first report, I am confident that the protocol of Zawadzki is useless and insecure, and I could show it in a page or so.  The authors adopt a very complicated path, which is not necessary at all, but I respect it. However, the authors should clarify the issues mentioned above, so that the manuscript is not misleading and the actual contribution of this work in the field is clear. 

Author Response

We thank Reviewer 1 for his comments, opinions and the effort put in our work. Our mentions to QKD in the introduction were meant to highlight the importance of entity authentication in the context of QKD, even if it is implicitly achieved through message authentication. However, after considering discussion with Reviewer 1, we agree that this could be confusing for readers and we have decided to remove the references relating QKD to QIA protocols to emphasize the fact that this is a work about the latter.   In doing this we have reworked the introduction beginning above "Our Contribution," specifically adding the lines 68-71 to make it abundantly clear we are not working directly with QKD but a different area of quantum cryptography.  Additionally small changes were made to the abstract to reflect these changes to the introduction.

Round 3

Reviewer 1 Report

The authors have made certain changes in their manuscript in order to avoid misunderstandings pertaining to entity authentication (identification), message authentication and QKD. The revised manuscript is certainly less misleading than before in this respect, although in the authors’ response to my previous report, there seems to exist still a confusion about the different protocols, and their use.  In any case, I still believe that this is a low-quality work, as is the work of Zawadzki. None of them is expected to have any impact in the field. It is obvious that the protocol of Zawadzki is insecure and useless, and this can be shown in a page or so. I do not understand why the authors have chosen a different rather complicated path, which does not offer anything. I also believe that it would have been better for the particular manuscript, if it had been submitted to the same journal as the work of Zawadzki. Given, however, that the editors of “Entropy” decided to accept the submission of the manuscript, and the other two referees do not see any problem with the quality of the work, the manuscript can be published  in “Entropy”, after the following changes.

Having clarified that the main topic of the work is entity authentication (identification) and not message authentication, there is a number of papers that have to be cited before its publication, as they are directly related to quantum entity authentication (identification):

https://arxiv.org/abs/2006.04522

https://www.mdpi.com/2410-387X/3/4/25

https://journals.aps.org/pra/abstract/10.1103/PhysRevA.101.042337

https://journals.aps.org/pra/abstract/10.1103/PhysRevA.97.012324

https://www.nature.com/articles/srep46047

https://link.springer.com/article/10.1007/s11128-018-1927-5

There may be more papers that I cannot remember now. In most of these protocols, entity authentication is achieved by means of a physical unclonable key, and its interaction with quantum states. 

Author Response

Thank you for the review and the list of resources, we have included these along with a sentence mentioning the broader field of study on lines 35 and 36.